# The Nutritional Profile of Food Advertising for School-Aged Children via Television: A Longitudinal Approach

**DOI:** 10.3390/children7110230

**Published:** 2020-11-17

**Authors:** Daniel Campos, Mireia Escudero-Marín, Camila M. Snitman, Francisco J. Torres-Espínola, Hatim Azaryah, Andrés Catena, Cristina Campoy

**Affiliations:** 1Department of Paediatrics, School of Medicine, University of Granada, Avda. Investigación 11, 18016 Granada, Spain; m.escuderomarin@gmail.com (M.E.-M.); fjtespinola@yahoo.es (F.J.T.-E.); rifappstudio@gmail.com (H.A.); 2EURISTIKOS Excellence Centre for Paediatric Research, Biomedical Research Centre, University of Granada, 18016 Granada, Spain; camisnitman@gmail.com; 3Mind, Brain and Behaviour International Research Centre (CIMCYC), University of Granada, 18011 Granada, Spain; acatena@ugr.es; 4Instituto de Investigación Biosanitaria de Granada (Ibs-GRANADA), Health Sciences Technological Park, 18012 Granada, Spain; 5Spanish Network of Biomedical Research in Epidemiology and Public Health (CIBERESP), Granada’s Node, Institute of Health Carlos III, 28029 Madrid, Spain

**Keywords:** screen time, childhood obesity, food preferences

## Abstract

The prevalence of childhood obesity continues to increase. Screen time, one of the most documented reasons for the obesogenic environment, enhances childhood obesity, since advertisements for unhealthy food products are still broadcast on channels for children. This is presently one of the main challenges for the government in Spain, since the current laws and obligations are not updated. This study aims to analyze food advertising aimed at children on Spanish television in 2013 and 2018 on children’s and general channels to test the effect of laws and obligations over time. In total, we viewed 512 h of the most viewed channels, two children’s and two general channels, during the week and on weekends during specific periods of 2013 and 2018. Food advertising was categorized as core, non-core, and other food advertisement (CFA, NCFA, and OFA, respectively) according to the nutritional profile. A total of 2935 adverts were analyzed, 1263 in 2013 and 1672 in 2018. A higher proportion of NCFAs were broadcast on children’s channels than in prior years, rising from 52.2% to 69.8% (*p* < 0.001). Nowadays, the risk of watching NCFAs on children’s channels compared to general channels turns out to be higher (Odds ratio > 2.5; *p* < 0.001), due to exposure to adverts for high-sugar and high-fat foods such as cakes, muffins, cookies, and fried and frozen meals rich in fat. In conclusion, the trends of nutritional profiles in food advertising on television are worsening over time, since the prevalence of NCFAs was higher in 2018 than in 2013. Currently, CFAs are not mainly broadcast on children’s channels, confirming high-risk exposure to non-core food advertising by watching them. Thus, food advertising laws and obligations should be adapted to increase compliance.

## 1. Introduction

The prevalence of childhood obesity increased in Europe from 1975 to 2016 [1,2]. In Spain, around 40% of children aged 3–8 years are overweight, and probably will become obese later in life [3]. This follows from the knowledge that screen time (ST) enhances childhood obesity by reinforcing unhealthy habits such as being sedentary when watching TV and posing a higher risk of exposure to advertisements for unhealthy food, and high energy intake while watching TV [4,5] is one of the most documented reasons for the obesogenic environment [6,7,8].

### 1.1. Consumer and Food Socialization in Childhood

Television, as an excellent means of mass communication, is considered as an instrument of socialization, a modeler of consciences, and an instigator of public behavior [9]. Currently, school-aged children spend more time watching television than the internet both during school days and on weekends [10]. Accordingly, TV food advertising has increased over time. It is a fact that marketing companies know that many children make decisions after being seduced by features in food adverts [11], and it is also known that food marketing influences people’s preferences for consuming healthy or non-healthy foods [12]. For example, junk food companies try to get children’s attention through rewards not related to the nutritional content of food [13]. Additionally, it is common to see adverts for high-sugar foods during children’s viewing times at school time compared to holidays [14]. Recent studies show that reducing TV time protects against childhood obesity at 7 years [15], while exposure to advertisements for high-energy food determines the energy balance, leading children to become overweight [16].

### 1.2. Food Advertising Influence and Literacy

Food advertising influences children’s attitudes and consumption preferences [12]. Emotional development is considered as an important determinant of human behavior [17]. Some neuromarketing studies associate the neural-brain network with emotions [18,19]. It is known that the duration and frequency of exposure to food advertising predicts final decision making [20,21]. Food companies focus on sugary products, since it helps them gain customer loyalty by stimulating the reward-processing circuitry of the brain in a similar pattern to addictive drugs [22], disabling the response to satiety [23,24]. For example, fast food is developed in very clever ways to make it addictive and very difficult to stop eating [25]. Furthermore, a rapid and continued drop in sugar intake is associated with similar symptoms as tobacco cessation [26]. A similar study [27] found that exposure to images of food like chocolate has a higher physiological effect. As children are underdeveloped and relatively inexperienced as consumers, they are more susceptible to advertising influence compared to adults [28]. Children are best approached though traditional media, like cartoons [29], and consequently, a large body of research has investigated children´s susceptibility to persuasive food advertising in TV [30,31,32,33]. Food advertising literacy is an effective intervention to cope with the unhealthy effects of food advertising [34]. Food-focused advertising literacy increases children’s knowledge of nutrition [35], then, many schools teach children how to cope with these commercials from the age of eight [36]. Additionally, individual socio-economic status influences food literacy skills, thus people suffering from social inequalities are more likely to show limited food literacy [37].

### 1.3. Regulatory Laws and WHO Recommendations

The current Spanish public health law was passed in 2011. It aims to protect children against unhealthy food advertising, concentrating efforts on reducing childhood obesity and enhancing healthy and active lifestyles, including moderate ST and regular family meal times, and recommends regular exercise to support physical and psychological health [38]. Early life habits are an accurate target for prevention, and this is presently one of the main challenges for the government in Spain, which has worked to implement a law for childhood obesity, which can be summarized in four points: (1) Reforming the Code for Regulation of the Advertising of Food and Beverages (PAOS code) for advertising self-regulation; (2) reducing added sugars; (3) using nutritional labelling following the Nutri-Core model [39]; and (4) updating the latest law (Spanish law 17/2011 for food safety) to ban unhealthy food and beverages in schools, as well as in hospitals, primary healthcare locations, and health administrations.

The development of pledges by the food industry is expanding around the world. The PAOS code, directed to children in Spain, is part of a national strategy for promoting healthy nutrition and physical activity to prevent obesity and enhance children´s healthy habits. It is a self-regulating code for food advertising in Spain created by companies in 2005 that unfortunately has not been modified since 2012 [40]. Some limitations and inconsistencies have been found, such as the choice of whether or not to participate and the self-regulating by companies [41].

The European Commission addressed the long-term effectiveness of such strategies, evolving a process for monitoring the extent and nature of food advertising to children and its regulation, and amending the EU’s Television Without Frontiers Directive to protect children from advertisements for unhealthy food when they are watching TV [42].

The World Health Organization recommends no more than 1 h of sedentary ST for children aged 2–12 years, and never at mealtimes [43]. Recent studies show an increase in advertising for fast foods and high-sugar foods in recent years [44,45,46]. Additionally, on both children’s and general channels, children are at high risk of watching adverts for unhealthy foods, which explains the high prevalence of childhood obesity [16].

Recent studies on TV advertising of food for children in Spain were focused on evaluating the latest approved laws and checking for compliance, finding significant increases in unhealthy food shown to children [47,48,49,50,51]. Consequently, some products, such as child-targeted packaged foods, are changing according to the latest obligations, although these changes are not enough [52]. Given the limitations of self-regulatory agreements for food advertising and their consequences to public health [53], a more evidence-based longitudinal approach is needed to highlight them.

Following the effort by the government to regulate this issue, our study aims to confirm a healthier trend in the nutritional profiles of food adverts shown on children’s channels compared to previous years, and to confirm a low risk of exposure to non-core food advertising on children’s and general channels. If these are confirmed, it shows that the long-term effects of the latest Spanish laws are contributing to address this public concern. Hence, this study investigated the nutritional profiles of food advertising on Spanish TV from 2013 to 2018 for the following purposes: (1) To compare the prevalence of non-core food advertising in 2013 against 2018; (2) to check the current broadcasting of core and non-core advertising on children’s and general channels; and (3) to test the effect of the latest Spanish laws and obligations on restricting food advertising in the most-viewed frame for children.

## 2. Materials and Methods

Spanish TV broadcasts were analyzed for 2 time points, 2013 and 2018. This involves the time between the last publication of Spanish food advertising regulations and the imminent publication of the new law for childhood obesity. A total of 3152 adverts were recorded over 512 h. The 2 most-watched Spanish channels for children (Disney Channel and Boing) and channels for all ages (Antena 3 and Telecinco), which allow advertising, were assessed. The period of the study was 2 months (April and May) for both 2013 and 2018. They were recorded on 4 consecutive days (2 weekdays and 2 weekend days, from 06:00 to 22:00). Food advertising broadcast from 06:00 to 07:00 was defined as outside the timeline or not analyzed since it is sleep time, thus that food advertising was not used. Consequently, after removing 217 adverts, the study comprised a total of 2935 adverts, 1263 in 2013 and 1672 in 2018. Adverts shown in children’s peak time slots (07:00–08:59 and 15:30–22:00 on weekdays, 07:30–10:29 and 15:30–22:00 on weekend days) and nonpeak time slots (09:00–15:29 on weekdays, 10:30–15:29 on weekend days) were also analyzed according to a previous study [50].

Each advertisement was analyzed and evaluated with predefined coding based on an inductive methodology (categorized from particular to general food and beverage features) according to the previous literature [49]. If possible, each advertisement was initially classified according to the nutritional composition of the food by nutrition labelling and later was included in the 3 main categories of adverts (core, non-core, and other). In particular, criteria were related to the nutritional quantity and quality of the foods and beverages being advertised, healthy habits or references after checking each recipe (Table 1), and nutraceutical products advertised in the Open Food Facts—Spain food products database [54]. Finally, food adverts were recorded and categorized according to a methodology based on a validated coding system [13,49], as follows: Core food advertisement (CFA; (low in energy and nutrient-dense), non-core food advertisement (NCFA; high in energy, with an unbalanced energy profile), and other food advertisement (OFA; special food like vitamins, supplements and baby formulas, and supermarkets) (Appendix A). Healthy adverts were defined as those that promote physical activity or other healthy lifestyle habits [47,50], after checking the labels of the foods and beverages advertised.

The research staff involved two nutritionists previously trained in classifying food and beverage adverts, and informatics and statistical scientists were in charge of making the TV records and the statistical analysis, respectively. A previous test was done to check their training for analyzing food advertising, thus pairwise comparison was done to confirm the validity of the nutritional analysis. WMP v.11 (SRS Labs. Inc. Santa Ana, CA, USA, 2013) and Excel v.2016 software (Microsoft Building Redmond, WA, USA) were used in the analysis, and the statistical analysis was done with IBM SPSS (IBM Corp. Released 2011. IBM SPSS Statistics for Windows, Version 20.0. Armonk, NY, USA). All adverts were stored in an HDD model Ciga HD 2TB that was programmed to record for 4 days in the time slots described above.

### Statistical Analysis

The statistical analyses included descriptive analysis to show the frequencies and rates of all variables and bivariate analysis through a contingency chart with χ^2^ test to assess the relationship between frequencies and relative risk. For the analysis of the trend of nutritional profiles of food advertising (TNPFA) at different time points, first a study of normality was conducted, testing the non-normal distribution by the Kolmogorov–Smirnoff test, followed by the Mann–Whitney nonparametric U-test to compare medians and interquartile range of the nutrients at each time point, considering results significant at *p* < 0.05 in all analyses.

The medians for TNPFA at different time points were calculated according to the nutritional profiles of advertised foods, and the broadcast frequency was used a correction factor in this analysis as shown in the following equation:$$\mathrm{TNPFA}\;=\frac{\;\mathrm{Nutritional}\;\mathrm{contents}\;\mathrm{of}\;\mathrm{food}\;\mathrm{advertised}\;\left(g\;\mathrm{and}\;\mathrm{kcal}\right)\;}{100\;g\;\mathrm{of}\;\mathrm{food}}\;\times \;\;\mathrm{Broadcast}\;\mathrm{Frequency}$$

## 3. Results

A total of 2935 adverts were analyzed (1263 in 2013 and 1672 in 2018) (Appendix A). The food adverts in the two years were significantly different. In 2018, a higher proportion of NCFAs were broadcast on children’s channels than years ago, rising from 52.2% to 69.8% (*p* < 0.001). OFAs increased from 0.7% to 15.7% (*p* < 0.001). Moreover, a higher proportion of CFAs were seen on general channels, increasing from 23.7% to 29.7% (*p* < 0.001), and OFAs decreased from 19% to 12.4% (*p* < 0.001). Similar trends were found for NCFAs on general channels (Figure 1).

A comparative analysis over time showed that adverts for fruit products without added sugar were not shown on children’s channels during 2018, while in 2013 their proportion was 30/579 (*p* < 0.001). Similarly, adverts with a higher content of fried foods were shown more in 2018 than 2013 (33/779 vs. 7/579; *p* < 0.001). Adverts for cakes, muffins, cookies, high-fat crackers, pies, and pastries were shown more on children’s channels than general channels in 2018 (40/779 vs. 13/893), contrary to 2013 (1/579 vs. 55/684; *p* < 0.001). Adverts showing healthy habits were not broadcast in different proportions on children’s channels and general channels (14/779 vs. 23/893; *p* < 0.001) (Table 1). Adverts for sugary drinks, high-sugar cereals, and chocolate followed a worse trend on general channels (*p* < 0.001).

The nutritional profile of NCFAs in 2018 was poorer than years ago, since the foods were higher in energy, carbohydrates, and sugar (*p* < 0.001). CFAs had a better nutritional profile in 2018 than 2013 (*p* < 0.001) (Table 2).

The analysis of advertising on children’s channels over time shows a higher risk of exposure to NCFAs, such as adverts for ice cream, high-fat meats, cakes, high-sugar cereals, and chocolate (*p* < 0.001) in 2018 than 2013 (Appendix A); conversely, adverts for healthy habits show a higher risk of exposure on children’s channels in 2013 than 2018 (Relative risk (RR) = 4.901; Confidence interval (CI): 2.740–8.767; *p* < 0.001; data not shown). In addition, the results show that watching children’s channels posed a high risk of exposure to adverts for high-sugar and high-fat foods like cakes, muffins, cookies, and breaded and frozen meals rich in fat (*p* < 0.001), but was protective for exposure to fast food meals and restaurants (Appendix A). Otherwise, there was a higher risk of exposure to adverts for sugar-sweetened drinks when watching general channels (Odds ratio (OR) = 34.894; CI = 5.004–266.011; *p* < 0.001; data not shown).

## 4. Discussion

This study used a longitudinal approach to observe the trend of nutritional profiles and the content of food advertising for children on Spanish TV, contributing to the current knowledge in this field. Overall, our results suggest that the nutritional quality of the food in adverts and the spread of unhealthy food advertising on TV channels for children are worsening over time. This association is explained by the higher risk of exposure to NCFAs on children’s channels than years ago, compared to general channels. Despite the latest improvements in laws against these activities, advertising of unhealthy foods has become more aggressive in both frequency and focus, in agreement with other studies [55]. Contrary to our review [44,45,46], the results show a low risk of exposure to fast food adverts on children’s channels, but there is still a high risk of exposure to adverts for high-sugar and high-fat foods, as concluded in previous research [14,22]. Moreover, health-oriented adverts such as for physical activity or healthy eating plans do not often appear on children’s channels; their trend decreased compared to 2013. This confirms that new actions should be developed since current regulations are not adequate, in line with what other authors have proposed [56]. Thus, there is a chance to develop effective social responsibility programs for sugary beverage advertising by the food industry and the governments [57].

Given the WHO’s recommendations for ST in 2019, it is worrying that 50% of children exceed these recommendations [58]. Furthermore, it is expected that sedentary activities and screen time are being extended according to social distancing guidelines during the COVID-19 pandemic [59], which is worrying since exposure to NCFAs is increasing. In light of this, a new problem appears related to children’s ST with general channels, since the results in this study show that there is a high risk of exposure to fast food advertising when watching these channels.

Recent studies using electroencephalographic techniques found an association between exposure to food pictures and decision making [60]. Additionally, eye-tracking was recently used in several studies in different areas like health, tourism, technology, and computing, finding important applications [61,62,63,64]. A recent study emphasized developing strategies to strengthen children’s coping skills and dispositional (associative network consisting of cognitive, moral, and affective beliefs) and situational (actual recognition of and critical reflection) attributions related to advertising [65]. Consequently, new technological applications are focused on getting children’s attention through mobiles, video games, and social networks, among others, and creating a working platform for prevention programs to control food advertising. Currently, food companies are developing apps based on nutritional databases to categorize food, such as more or less adequate, and this path provides an opportunity to promote nutritional education for consumers to be able to effectively judge food advertising. This is a great opportunity to standardize and check such databases and apps by law, hence it could help to control food adverts and improve nutritional education for children and parents.

### 4.1. Policy Implications

Prevention of overweight and obesity is a key target in contemporary health care [66]. Food advertising and childhood health are among the biggest concerns for the government, since food advertising shown on TV is imbalanced and could be promoting a high risk of obesity [67]. Consequently, TV food advertising should be involved in the new TV spaces that consider a sequence of health interventions such as developing programs in high-quality contents. Additionally, easy and clear messages should be promoted, and adequate food advertising literacy that protects children into the evidence-based approach to increase the effectiveness of the strategies should be encouraged. It must also allow a mobilized society, with the necessary practicality and feasibility, to change the government and the food industry, since their current positions are inappropriate [68,69].

Until now, not enough has been done to ensure that children are protected from high exposure to unhealthy food advertising. Our results provide arguments to improve the law for food advertising to children. For example, the main policy implication goes through the development of new pledges, since the current state of pledges by the food industry suggests improvement through mandatory participation or laws for reducing exposure and re-addressing the power of food advertising [49]. Moreover, strategies to prevent childhood obesity related to TV advertising should use an evidence-based approach in order to succeed. For example, the effectiveness of “protective” messages in food advertising for children is under question because the results are not as expected [70]. This may be because healthy messages shown in food advertisements on TV do not usually receive attention, since they always appear in illegible type for only a short time. This suggests focusing on the programs most watched by children, taking actions that prioritize health concerns above all else, such as developing educational programs on nutritional food and healthy habits, having four daily meals, portion sizes, dairy product consumption, fruits and vegetables, water, and the importance of physical activity, to address the problem [71,72,73,74]. Likewise, interventions should be supervised by governments, since the nutritional education provided by food companies is considered misleading and inconsistent with the appropriate advice [75,76]. The results of this study provide new knowledge in line with the finding of other European projects, to promote the adoption of a commonly agreed European Union definition of ‘unhealthy’ food [46].

The interest of the food industry in the management of pledges for advertising is confirmed [77] by an increase in the number of TV adverts compared to years ago, as the results of our study show. The government of Spain continues to support self-regulation following the PAOS code [40,52]. The major limitations of the current pledges are the self-regulation and the voluntary involvement by companies, showing that these pledges are not effective since they represent a risk of not protecting children against the harmful effects of food advertising. For example, self-regulation is intended to redefine “child-directed advertising” to reduce television advertising of unhealthy foods [78]. The results in this study are in line with the World Health Organization’s statement that “self-regulatory or voluntary schemes often have a narrow scope, weak criteria and limited government oversight” [79], which highlights the ethical and political issues that arise when important food and beverage companies continue to exploit the policy space created. For example, the Coca-Cola Company has been criticized for promoting physical activity to distract attention from its products [76]. According to a study developed at King’s College London, the lack of commitment to this target is due to the idea of corporate social responsibility in the age of obesity, since companies are selling health as “brand value,” shifting the blame from food to diet and from diet to sedentariness, proving a lack of responsibility. Thus, obesity prevention has become a veritable industry in itself managed by the food and drink industry [80].

Other specific actions include developing educational programs for children, but families and institutions should also be included in the regulatory guidelines [81]; this is an opportunity to redirect the use of tech devices towards more healthy practices being offered. Parents, food companies, governments, and other institutions must join efforts on a large scale to favor the behavioral challenge for food advertisement literacy [73,82], since the legal regulations can potentially make positive cost-effective intervention when television advertising of high-fat, high-sugar, and high-salt foods and beverages to children is restricted [83,84]. Consumers, legislators, and the media must be informed about unfair approaches used by food and beverage companies and fast food chains, since the obesogenic environment continues to expand [85], just as food advertising literacy should be considered for them [36]

Consequently, governments have to recommend “healthy” TV viewing habits, such as by notifying parents about the potential for TV to displace time spent in physical activity and advising them to not allow their children to watch TV or devices during meals [86]. Children who have high physical activity levels, low screen time (ST), and healthy eating habits are at lower risk of becoming obese adults, thus they must be the examples to prevent the obesogenic environment [87], so TV advertising should at least consider running adverts for healthy habits like engaging in physical activity and eating healthy food during peak slots for children.

### 4.2. Limitations and Strengths

Some perceived limitations of this study could be related to the lack of interesting advert features, as the context of the programs on which the adverts were broadcast, the time they were broadcast, and their “protective” message, as well as the lack of covariates related to food choices (well-known children’s characters, athletes, movie celebrities, gifts and discounts for purchasing products) were not analyzed. Another limitation is the relatively small period of analysis (only three months per year); because advertising during the summer months was not analyzed, the potential number of adverts could be different, since both children’s schedules and advert contents would be different on holidays. Besides, since this is not an intervention study, the children’s perceptions after viewing food advertisements were not evaluated, which could be a key point to consider for future studies. The main strength of our study is to analyze current food advertising compared with advertising from years ago, under the same methods. Additionally, the results in this study demand deep research into the utility of literacy interventions in food advertising for the promotion of healthy eating. As such, the future policy guidelines should attend to the follows main points related to food advertising literacy: (1) Need to develop a healthy advertising disclosure in new TV spaces, (2) increasing the general knowledge about healthy and unhealthy food advertising during childhood and developing school interventions to children on how to cope with unhealthy food advertising, (3) increasing parent’s advertising literacy, (4) increasing awareness among food companies and advertising professionals, and (5) increasing citizen awareness of complaint mechanisms.

## 5. Conclusions

The trend of nutritional profiles in food advertising on Spanish television for children has worsened over the years. CFAs were mainly not broadcast on channels for children, while NCFAs were broadcast more in 2018 than 2013. Currently, watching children’s channels is a risk factor for exposure to NCFAs. Thus, food advertising regulations should be adopted to increase compliance.

## Figures and Tables

**Figure 1 children-07-00230-f001:**
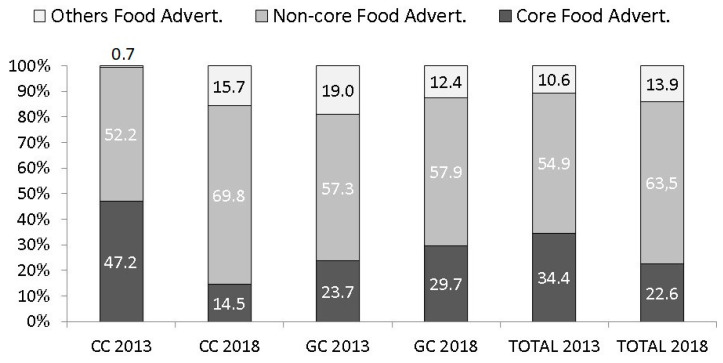
Percentage of TV adverts broadcast on Spanish children’s channels (CC) [6] and general channels (GC) over time.

**Table 1 children-07-00230-t001:** Proportions and frequencies of food advertising broadcast on Spanish TV in 2013 and 2018 by type of channel.

Advertisement Subtypes	2013		2018		Total	
	Children’s Channels *n* (%)	General Channels *n* (%)	Children’s Channels *n* (%)	General Channels *n* (%)	2013 *n* (%)	2018 *n* (%)
Core Food Advertisements (CFAs)						
Vegetables and vegetable products without added sugar	0 (0.0) ^c^	0 (0.0) ^c^	0 (0.0) ^c^	5 (0.6) ^d^	0 (0.0) ^a^	5 (0.3) ^a^
Bottled water	1 (0.2) ^c^	9 (1.3) ^c^	0 (0.0) ^c^	6 (0.7) ^c^	10 (0.8) ^a^	6 (0.4) ^a^
Dairy products	0 (0.0) ^c^	57 (8.3) ^c^	0 (0.0) ^d^	1 (0.1) ^d^	57 (4.5) ^a^	1(0.1) ^d^
Low-fat/reduced-fat milk, yoghurt, custard, cheese, alternatives (including probiotic drinks), infant food (excluding milk formulas)	191 (33.0) ^c^	0 (0.0) ^c^	96 (12.3) ^d^	113 (12.7) ^d^	191 (15.1) ^a^	209 (12.5) ^d^
Fruit and fruit products without added sugar	30 (5.2) ^c^	12 (1.8) ^c^	0 (0.0) ^d^	15 (1.7) ^c^	42 (3.3) ^a^	15 (0.9) ^d^
Meat and similar products	0 (0.0) ^c^	22 (3.2) ^c^	2 (0.3) ^c^	17 (1.9) ^c^	22 (1.7) ^a^	19 (1.1) ^a^
Soups, salads, and sandwiches, frozen meals, and low-fat savory sauces	0 (0.0) ^c^	22 (3.2) ^c^	1 (0.1) ^c^	51 (5.7) ^d^	22 (1.7) ^a^	52 (3.1) ^d^
Bread (including high-fiber bread and low-fat crackers), rice, pasta, and noodles	0 (0.0) ^c^	32 (4.7) ^c^	0 (0.0) ^c^	34 (3.8) ^c^	32 (2.5) ^a^	34 (2.0) ^a^
Healthy habits	51 (8.8) ^c^	8 (1.2) ^c^	14 (1.8) ^d^	23 (2.6) ^d^	59 (4.7) ^a^	37 (2.2) ^d^
TOTAL (CFA), *n* (%)	273 (47.2) ^a,c^	162 (23.7) ^a,d^	113 (14.5) ^b,c^	265 (29.7) ^b,d^	435 (34.4) ^a^	378 (22.6) ^b^
Non-core Food Advertisements (NCFAs)						
Frozen/fried potato products	0 (0.0) ^c^	0 (0.0) ^c^	0 (0.0) ^c^	0 (0.0) ^c^	0 (0.0) ^a^	0 (0.0) ^a^
Ice cream and iced confections	42 (7.3) ^c^	0 (0.0) ^c^	1 (0.1) ^d^	0 (0.0) ^c^	42 (3.3) ^a^	1 (0.1) ^d^
Fruit juice and fruit drinks	7 (1.2) ^c^	15 (2.2) ^c^	0 (0.0) ^c^	0 (0.0) ^d^	22 (1.7) ^a^	0 (0.0) ^d^
Breaded or battered meat and similar products and high-fat frozen meals	7 (1.2) ^c^	7 (1.0) ^c^	33 (4.2) ^d^	10 (1.1) ^c^	14 (1.1) ^a^	43 (2.6) ^d^
Alcohol	0 (0.0) ^c^	1 (0.1) ^c^	0 (0.0) ^c^	12 (1.3) ^d^	1 (0.1) ^a^	12 (0.7) ^d^
Cakes, muffins, cookies, high-fat crackers, pies and pastries	1 (0.2) ^c^	55 (8.0) ^c^	40 (5.1) ^d^	13 (1.5) ^d^	56 (4.4) ^a^	53 (3.2) ^a^
Sugar-sweetened drinks, including soft drinks, cordials, sports drinks, and flavor additions	0 (0.0) ^c^	17 (2.5) ^c^	0 (0.0) ^c^	40 (4.5) ^d^	17 (1.3) ^a^	40 (2.4) ^d^
Snack foods, including chips, extruded snacks, popcorn, snack and granola bars, sugar-sweetened fruit and vegetable products, and sugar-coated or salted nuts	14 (2.4) ^c^	40 (5.8) ^c^	19 (2.4) ^c^	26 (2.9) ^d^	54 (4.3) ^a^	45 (2.7) ^d^
High-sugar or low-fiber breakfast cereals	59 (10.2) ^c^	23 (3.4) ^c^	166 (21.3) ^d^	85 (9.5) ^d^	82 (6.5) ^a^	251 (15.0) ^d^
Whole milk, yoghurt, custard, dairy desserts, cheese and similar products	136 (23.5) ^c^	71 (10.4) ^c^	152 (19.5) ^c^	72 (8.1) ^c^	207 (16.4) ^a^	224 (13.4) ^d^
High-fat, high-sugar, high-salt spreads, oils, and high-fat savory sauces	5 (0.9) ^c^	31 (4.5) ^c^	0 (0.0) ^d^	13 (1.5) ^d^	36 (2.9) ^a^	13 (0.8) ^d^
Chocolate and confectionery	5 (0.9) ^c^	60 (8.8) ^c^	108 (13.9) ^d^	145 (16.2) ^d^	65 (5.1) ^a^	253 (15.1) ^d^
Fast-food restaurants or meals	26 (4.5) ^c^	72(10.5) ^c^	25 (3.2) ^c^	101 (11.3) ^c^	98 (7.8) ^a^	126 (7.5) ^a^
TOTAL (NCFA), *n* (%)	302 (52.2) ^a,c^	392 (57.3) ^a,c^	544 (69.8) ^b,c^	517 (57.9) ^a,d^	694 (54.9) ^a^	1061 (63.5) ^b^
Other Food Advertisements (OFAs)						
Baby and toddler milk formulas	4 (0.7) ^c^	2 (0.3) ^c^	107 (13.7) ^d^	0 (0.0) ^c^	6 (0.5) ^a^	107 (6.4) ^d^
Vitamin and mineral supplements	0 (0.0) ^c^	73 (10.7) ^c^	0 (0.0) ^c^	28 (3.1) ^d^	73 (5.8) ^a^	28 (1.7) ^d^
Supermarkets that advertise mostly core food	0 (0.0) ^c^	40 (5.8) ^c^	1 (0.1) ^c^	30 (3.4) ^d^	40 (3.2) ^a^	31 (1.9) ^d^
Tea and coffee	0 (0.0) ^c^	0 (0.0) ^c^	0 (0.0) ^c^	48 (5.4) ^d^	0 (0.0) ^a^	48 (2.9) ^d^
Supermarkets that advertise mostly non-core food	0 (0.0) ^c^	0 (0.0) ^c^	0 (0.0) ^c^	5 (0.6) ^d^	0 (0.0) ^a^	5 (0.3) ^a^
Supermarkets with no specified food	0 (0.0) ^c^	15 (2.2) ^c^	14 (1.8) ^d^	0 (0.0) ^d^	15 (1.2) ^a^	14 (0.8) ^a^
TOTAL (OFA), *n* (%)	4 (0.7) ^a,c^	130 (19.0) ^a,d^	122 (15.7) ^b,c^	111 (12.4) ^b,c^	134 (10.6) ^a^	233 (13.9) ^b^
Total Food Advertisements, *n* (%)	579 (100)	684 (100)	779 (100)	893 (100)	1263 (100)	1672 (100)

^a,b^ indicate significant differences in proportions of adverts shown in 2013 vs. 2018 for the same channel at significance level *p* < 0.001. ^c,d^ indicate significant differences in proportions of adverts shown between channels for the same year at significance level *p* < 0.001.

**Table 2 children-07-00230-t002:** Trend of nutritional profiles of food advertising (TNPFA) shown on children’s channels during the period of the study.

Nutritional Information	CFA					NCFA				
	2013 (*n* = 192)		2018 (*n* = 118)		*p*	2013 (*n* = 302)		2018 (*n* = 599)		*p*
	X	IQR	X	IQR		X	IQR	X	IQR	
Energy (kcal)	110.0	34.0–110.0	64.0	62.0–72.0	<0.001	293.0	286.0–379.0	383.0	293.0–472.0	<0.001
Lipids (g)	2.7	0.4–2.7	1.0	0.8–2.8	ns	16.0	3.5–22.0	18.0	12.0–23.0	ns
Saturated lipids (g)	1.7	0–1.7	0.6	0.5–1.2	<0.001	7.5	1.5–15.5	5.0	3.2–11.0	ns
Carbohydrates (g)	14.4	4.0–14.4	8.9	6.7–13.0	<0.001	24.0	17.4–64.6	54.0	3.1–70.0	<0.001
Sugars (g)	13.5	3.8–13.5	8.9	6.7–12.0	<0.001	14.9	4.5–23.0	18.0	0.5–56.0	<0.001
Proteins (g)	6.9	2.9–6.4	2.8	2.6–3.1	<0.001	6.8	4.4–11	7.5	5.0–15.0	ns

X, median; IQR, interquartile range; ns, non-significant. Significance level for statistical tests: *p* < 0.05. All nutrient contents refer to 100 g of food. Advertisements without labels were excluded from the statistical analysis.

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
