# Peer review of "The Nutritional Profile of Food Advertising for School-Aged Children via Television: A Longitudinal Approach"

_children, 2020, doi:10.3390/children7110230_

Round 1

Reviewer 1 Report

I am pleased to see that previous recommendations have for the most part been dealt with. I would like to make the following comments on the re-submitted manuscript:

Section 1.2 Food advertising influence and literacy

The concept of advertising literacy needs some further unpacking to clarify its importance within the study. This would assist in expanding upon the arguments relating to food advertising literacy on lines 288 and, in particular, 312-313.

The concept of ‘children’

It is still not clear what is meant by ‘children’ in terms of an age range. Could the viewership of Disney and Boing be used to settle this?

4.1 Policy implications

Line 244: What is meant by ‘Spanish advertising’ more specifically? Likewise, ‘a sequence of health interventions’ needs clarifying.

General

Line 241: Consider inserting a full stop after [62].

Line 284: Others specific action…

Line 308-311: This sentence needs addressing.

Author Response

Response to Reviewer 1 Comments

Point 1:

Section 1.2 Food advertising influence and literacy

The concept of advertising literacy needs some further unpacking to clarify its importance within the study. This would assist in expanding upon the arguments relating to food advertising literacy on lines 288 and, in particular, 312-313.

Response 1: The concept of advertising literacy has been developed, specifically, the concept has been revised and new ideas have been included in lines 71-80.

Specifically, new literature has been included in the review, such as:

  • Zarouali B, De Pauw P, Ponnet K, Walrave M, Poels K, Cauberghe V, Hudders L: Considering children’s advertising literacy from a methodological point of view: Past practices and future recommendations. Journal of Current Issues & Research in Advertising 2019, 40:196-213.
  • Daems K, Moons I, De Pelsmacker P: Co-creating advertising literacy awareness campaigns for minors. Young Consumers 2017.
  • Vanwesenbeeck I, Wolf D, Lambrecht I, Hudders L, Cauberghe V, Adams B, Lissens S: Minors’ advertising literacy in relation to new advertising formats: identification and assessment of the risks. AdLit SBO; 2016.
  • Palumbo R, Adinolfi P, Annarumma C, Catinello G, Tonelli M, Troiano E, Vezzosi S, Manna R: Unravelling the food literacy puzzle: Evidence from Italy. Food Policy 2019, 83:104-115.

Also, the discussion section has benefited from this point, since these ideas have been contrasted in lines 300-301 and 321-330. Besides, in the first paragraph of the section “Policy implications” in the discussion, also it has been disused this point.

Point 2:

The concept of ‘children’

It is still not clear what is meant by ‘children’ in terms of an age range. Could the viewership of Disney and Boing be used to settle this?

Response 2: In the second paragraph of the introduction section (line 51), the term "children" has been detailed as "school-aged children", according to the title section. We have deeply discussed this point and we decided not to apply this change in the whole document because it could be the cause of less fluent reading.

Point 3:

4.1 Policy implications

Line 244: What is meant by ‘Spanish advertising’ more specifically? Likewise, ‘a sequence of health interventions’ needs clarifying.

Response 3: This message has been modified at your suggestion, to make it clearer and more specific. We have changed the term "Spanish advertising" with "TV food advertising". Also, the whole of the message has been detailed (lines 250-254)

Point 4:

General

Line 241: Consider inserting a full stop after [62].

Line 284: Others specific action…

Line 308-311: This sentence needs addressing.

Response 4: All general points have been changed according to the reviewer.

Consider inserting a full stop after [62]. à Line 247 (Now is the reference 66)

Others specific action… à Line 392

This sentence needs addressing à The lines have been addressed (Now is in lines 327-324)

Reviewer 2 Report

I have already reviewed this article, and I see that the suggestions I made are mostly collected.

Author Response

Response to Reviewer 2 Comments

Point 1: I have already reviewed this article, and I see that the suggestions I made are mostly collected.

Response 1: Changes were made according to the reviewer's suggestions in the previous round

This manuscript is a resubmission of an earlier submission. The following is a list of the peer review reports and author responses from that submission.

Round 1

Reviewer 1 Report

The subject of the article is very interesting, because we need research to help us prevent childhood obesity, which is a very serious health problem. I believe that the approach of the work, as a whole, is adequate, but could be improved if it incorporated information on the following aspects: 1.- In the induction, three objectives are offered, but the authors' research hypotheses are not indicated. 2.- The methodology should explain the reasons for choosing those two specific years for the study and why an intermediate assessment was not made, for example in 2015. When analyzing two historical moments, and not a continuous evolution during the time period , the work loses a certain value of its longitudinal character. 3.- I think more information should be offered about the analysis and evaluation methodology of each advertisement, the training of judges, if a comparison was made between pairs that the evaluations were similar, etc. 4.- I think the absence of certain co-variables in the study should be better explained. For example, why the perception of children regarding advertisements has not been evaluated; Whether an ad has good or bad nutritional information is important, but perhaps the key is to know which ones are seen and which are not seen by children.

Reviewer 2 Report

Positioning of the paper:

The paper needs more robust anchoring. The literature review deals with decision-making and neuromarketing. Both aspects require further exploration to make them relevant if they are going to be used to discuss the findings and then make relevant policy recommendations. However, these aspects of theory have not been used for this purpose. The authors may wish to consider exploring the literatures associated with consumer socialisation, food socialisation and advertising literacy.

The title mentions ‘school-aged children’. Elsewhere in the paper there is a reference to adolescents. Much clearer parameters are needed in terms of age. A 14-year-old is very different from a seven-year-old in terms of their advertising literacy and choice of TV programmes.

General definitions:

Clear definitions of core- and non-core foods are required.

Legislative context:

A more complete, yet concise, account of the Spanish ‘law of childhood obesity’ is needed.

Likewise, the PAOS code needs to be spelled out and accounted for.

Given the global scope of the journal, references to approaches used in other countries would be helpful, especially as childhood obesity is a global problem and legislation is one aspect used by governments to try to tackle it.

Research methods:

The decision not to include adverts aired during the 06.00-07.00 time-slot was perplexing, yet the evening time-slot extended until 22.00. Perhaps this is a cultural aspect in Spain? In some other European countries, children tend to be early risers and will not be allowed to watch TV late at night. Here, again, the actual ages of the children are important. A five-year-old may be awake at 06.00, but will be in bed by perhaps 20.00. A 14-year-old may wake later, but perhaps watch TV until 22.00.

Which were the TV channels?

A clearer account of the inductive methodology is needed.

Presentation of findings:

The paper gives a thorough, quantitative account of the type of adverts.

Policy implications:

Given the approach of the paper, it is likely that this section would be expected to provide the original contribution. However, the discussion is frequently unclear. It is also not clear quite how these recommendations follow from the findings of the paper.

Specific policy recommendations are required here that are appropriate for different age groups of children. Given the global scope of the target journal, the discussion would also benefit from taking into account approaches taken in other European countries.

General comments:

Acronyms need spelling out at first use within the abstract.

The abstract needs to be more precise in terms of what is meant by laws and ‘commitments’. Overall, the abstract should give a concise, complete account of the paper.

The introduction is very brief and would benefit from at least an overview of the structure of the paper.

In several places, the meaning of sentences is unclear. For instance:

‘Recent studies show an increase in Fast foods and high sugar products in recent years’. (Lines 86-87) Is this in terms of advertising or consumption?

‘Consequently, some of the products such as child-targeted packaged foods are changing over time according to new commitments….’ (Lines 92-93) Changing in what way?

Other high sugar adverts of sugary drinks, high sugar cereals and chocolate follow a worse trend in Generalist channels. (Lines 164-165) In what way worse?

English language:

The English language needs attention. There are issues with sentence construction, syntax, expression and vocabulary.

Sentences tend to be too long at times. For instance, the sentence running from line 213-220.